# Favorable Conditions for the Detection of EGFR T790M Mutation Using Plasma Sample in Patients with Non-Small-Cell Lung Cancer

**DOI:** 10.3390/cancers15051445

**Published:** 2023-02-24

**Authors:** Insu Kim, Hee Yun Seol, Soo Han Kim, Mi-Hyun Kim, Min Ki Lee, Jung Seop Eom

**Affiliations:** 1Department of Internal Medicine, Dong-A University Hospital, Busan 49201, Republic of Korea; 2Department of Internal Medicine, Pusan National University Yangsan Hospital, Yangsan 50612, Republic of Korea; 3Department of Internal Medicine, Pusan National University School of Medicine, Busan 49241, Republic of Korea; 4Department of Internal Medicine, Pusan National University Hospital, Busan 49241, Republic of Korea; 5Biomedical Research Institute, Pusan National University Hospital, Busan 49241, Republic of Korea

**Keywords:** diagnostic molecular pathology, epidermal growth factor receptor, liquid biopsy, lung cancer, tyrosine kinase inhibitor

## Abstract

**Simple Summary:**

Up to 60% of patients with non-small-cell lung cancer with epidermal growth factor receptor (EGFR) acquire a T790M mutation after first-line EGFR-tyrosine kinase inhibitors (TKIs). Although a liquid biopsy using plasma samples for the detection of the T790M mutation is recommended initially, some patients receive tissue re-biopsy due to the possible false negative results of liquid biopsy. In this study, 40% of patients with one or two metastatic organs at the time of re-biopsy had false negative plasma sample results, and 69% of patients with three or more metastatic organs at the time of re-biopsy had positive plasma results. In multivariate analysis, three or more metastatic organs at the initial diagnosis were independently associated with detection of the T790M mutation using a plasma sample, which means the detection rate of the T790M mutation using a plasma sample was significantly increased in patients with more tumor burden.

**Abstract:**

Background: Detection of the epidermal growth factor receptor (EGFR) T790M mutation using plasma samples has been considered simple and non-invasive, but the relatively high false negative results lead to additional tissue sampling in some patients. Until now, the characteristics of patients who prefer liquid biopsy have not yet been established. Methods: To evaluate the favorable conditions for the detection of T790M mutations using plasma samples, a multicenter retrospective study was performed between May 2018 and December 2021. Patients whose T790M mutation was detected in a plasma sample were classified as the plasma positive group. Study subjects with a T790M mutation not detected in a plasma sample but only in a tissue sample were grouped as the plasma false negative group. Results: Plasma positive and plasma false negative groups were found in 74 and 32 patients, respectively. As a result, 40% of patients with one or two metastatic organs at the time of re-biopsy had false negative plasma sample results, and 69% of patients with three or more metastatic organs at the time of re-biopsy had positive plasma results. In multivariate analysis, three or more metastatic organs at initial diagnosis were independently associated with the detection of a T790M mutation using plasma samples. Conclusion: Our results demonstrated that the detection rate of a T790M mutation using plasma samples was related to the tumor burden, particularly to the number of metastatic organs.

## 1. Introduction

Lung cancer is the most common cancer and the leading cause of cancer-associated death in the world [1]. With the changes in the paradigm of non-small-cell lung cancer (NSCLC) treatment, there have been many advances in the overall survival and progression-free survival [2]. In particular, the development in diagnostic modalities of specific somatic mutations has broadened the selection of applicable treatment options [3]. Most importantly, advanced NSCLC patients who harbor epidermal growth factor receptor (EGFR) mutations, which were found in about half of the advanced lung adenocarcinoma cases in Asians, can be treated with EGFR-tyrosine kinase inhibitor (TKI) [4]. However, resistances to EGFR-TKI have emerged about 10–14 months after the initiation of EGFR-TKI treatment [5].

Various mechanisms of resistance to EGFR-TKI treatment have been reported, and a T790M mutation is known as the most common acquired resistance, occurring in up to 60% of the patients [6]. Fortunately, patients with a T790M mutation can be effectively treated with the third generation EGFR-TKI, such as lazertinib or osimertinib, with progression-free survival of 11–17 months [7,8]. However, in patients in whom the T790M mutation was not detected, progression-free survival was only 3.1 months [9]. Thus, significant efforts must be made to detect T790M mutations in patients whose disease has progressed after treatment with first- or second-generation EGFR-T KI. Therefore, verification of the presence or absence of a T790M mutation has been considered to be the most important process for determining the next treatment strategy after confirming resistance to the first-line EGFR-TKI treatment.

For the identification of T790M mutations, mutation analysis is performed using tissue or plasma samples, each with its own advantages and disadvantages. Although the sensitivity of tissue re-biopsy is known to be more accurate in the detection of the T790M mutation than that of plasma samples, there have been concerns about the invasiveness of procedures and the limitation in overcoming spatial heterogeneity [10,11]. On the contrary, detection of the T790M mutation using plasma samples, so-called liquid biopsy, is simple and convenient, though the relatively high false negative results lead to additional tissue sampling in some patients. So far, the characteristics of patients who prefer liquid biopsy have not yet been established. Therefore, we evaluated the favorable conditions for the detection of T790M mutations using plasma samples.

## 2. Materials and Methods

### 2.1. Study Population

Between May 2018 and December 2021, this multicenter, retrospective, and observational study was performed in accordance with the STROBE statement. The electronic medical records of the consecutive patients who received a liquid biopsy for the detection of the T790M mutation were reviewed at three university-affiliated tertiary referral hospitals as follows: Pusan National University Hospital; Pusan National University Yangsan Hospital; and Dong-A University Hospital in the Republic of Korea. The inclusion criteria for this study were a pathologically confirmed diagnosis of advanced NSCLC, the presence of an EGFR exon 19 or 21 mutation at initial diagnosis, disease progression after first-line systemic therapy with 1st- or 2nd-generation EGFR-TKIs (erlotinib, gefitinib, or afatinib), and availability of at least one liquid biopsy using plasma and/or tissue re-biopsy at the time of disease progression. Patients who had an exon 20 or T790M mutation at the first diagnosis or without a liquid biopsy after progression were excluded. Considering the nature of this retrospective study, the need for informed consent by the patients was waived.

### 2.2. Study Groups

Patients whose T790M mutation was detected in a plasma sample were classified as the plasma positive group. Study subjects with a T790M mutation not detected in a plasma sample but only in a tissue sample were grouped as the plasma false negative group.

### 2.3. Tissue Acquisition Methods

Tissue re-biopsy for the detection of the T790M mutation was performed in patients with negative results in plasma samples, or some patients received EGFR mutation tests simultaneously using both tissue and plasma samples. The tissue acquisition method was selected from one of the following procedures, depending on the location of the accessible lesion, applicability of the procedure, and the general medical condition of the patient: percutaneous needle aspiration or biopsy, transbronchial biopsy using a radial probe endobronchial ultrasound, endobronchial ultrasound-guided transbronchial needle aspiration, bronchoscopic forceps biopsy, ultrasound-guided biopsy of extrathoracic lesion, or video-assisted thoracoscopic surgery.

### 2.4. T790M-Resistant Mutation Analysis

Using tissue specimens, T790M mutational tests were performed using an EGFR Mutation Detection Kit (PNA clamp™; Panagene, Daejeon, Republic of Korea). In liquid biopsies, two methods were used, either an EGFR Mutation Detection Kit (PNA clamp™; Panagene, Daejeon, Republic of Korea) or a Cobas EGFR Mutation Test v2 (cobas^®^; Roche Molecular Systems, Pleasanton, CA, USA), on the plasma sample.

### 2.5. Statistical Analysis

Variables are described as numbers (%) or means (standard deviation (SD)), as appropriate. As a method for comparing two independent groups, the chi-squared or Fisher’s exact test was performed for categorical variables, and an independent t-test or Wilcoxon rank sum test was performed for continuous variables depending on the satisfaction of the normality. Univariate and multivariate logistic regression analyses were performed to find favorable clinical factors for the detection of the T790M mutation using plasma samples. A logistic regression model is used to present the odds ratio and 95% confidence interval for each variable. Statistical analyses were conducted using SPSS version 22.0 for Windows software (SPSS Inc., Chicago, IL, USA) and the R statistical program version 4.0.5 (GNU GPL). A *p*-value < 0.05 was considered significant.

## 3. Results

### 3.1. Study Participants

A total of 156 patients underwent liquid biopsies using plasma samples (Figure 1). A total of 308 re-biopsies (including plasma and tissue samples) were performed in 156 study subjects as follows: 64 (41%) underwent only one liquid biopsy, and 92 (59%) underwent two or more re-biopsies with tissue or plasma samples (an average of 2.6 biopsies per person). The overall detection rate of the T790M mutation was 68% (106 of 156 patients), and plasma positive and plasma false negative groups were found in 74 and 32 patients, respectively.

Baseline characteristics of plasma positive and plasma false negative groups are summarized in Table 1. The mean age was 67 years (SD, 10), and 54% were male. The Eastern Cooperative Oncology Group performance status was 0 for 42 individuals (57%), 1 for 28 individuals (38%), and ≥2 for 4 individuals (5%) in the plasma-positive group. There was no significant difference between the two groups when compared to the plasma false negative group (*p* = 0.768). The number of never-smokers was 47 (64%) in the plasma positive group and 18 (56%) in the plasma false negative group; no significant difference was observed between the two groups (*p* = 0.626). Regarding EGFR mutation status, 45 (60%) and 17 (53%) patients had the 19del mutation, and 29 (40%) and 15 (47%) patients had the L858R mutation in the plasma-positive and false negative groups, respectively. No significant difference was noted between the two groups. In the plasma positive group, the number of stage IV patients at initial diagnosis was significantly higher than that in the plasma false negative group (84% vs. 69%; *p* = 0.001).

### 3.2. EGFR Mutation Status on Secondary Biopsy

Real-time PCR (RT-PCR) and PNA clamping were used to detect EGFR mutations in secondary biopsies. In EGFR mutation tests using tissue and plasma in the plasma positive group, 46 cases in 42 patients (47%) had the 19del + T790M mutation, and 30 cases in 26 patients (31%) had the L858R + T790M mutation. In the plasma false-negative group, 18 patients (28%) had the wild type, 17 patients (27%) had the 19del + T790M mutation, and 15 patients (23%) had the L858R + T790M mutation (Table 2).

### 3.3. Treatment-Related Parameters

For the first-line EGFR-TKI treatment, gefitinib was used in 29 patients (28%), erlotinib in 12 (11%), and afatinib in 65 (61%) (Table 3). The mean duration of the first EGFR-TKI treatment was 21 months (SD, 17), and there was no statistical difference between the plasma positive and plasma false negative groups (19 months vs. 24 months, respectively; *p* = 0.179). The first and the best responses to the first-line EGFR-TKI were not significantly different between the two groups (*p* = 0.834 and *p* = 0.440, respectively).

### 3.4. Reassessment of Clinical Stage and Metastatic Organs

When the T790M mutation was found, re-staging was performed and compared between plasma positive and plasma false negative groups (Table 4). A significant difference was found in the number of metastatic organs between the plasma positive and plasma false negative groups (*p* = 0.028); the proportion of patients with three or more metastatic organs in the plasma positive and plasma false negative groups was 57% and 32%, respectively. Moreover, 40% of the patients with one or two metastatic organs at the time of re-biopsy had false negative results in the present study. Otherwise, there were no significant differences in the T, N, and M stages between the two study groups (*p* = 0.630, *p* = 0.685 and *p* = 0.239, respectively).

### 3.5. Favorable Factors for T790M Mutation Detection Using Plasma Samples

Univariate and multivariate logistic regression analyses were performed to identify the independent factors associated with the favorable conditions for T790M mutation detection using plasma samples (Table 5). Thus, three or more metastatic organs at the initial diagnosis were independently associated with T790M mutation detection using plasma samples (odds ratio, 0.014; 95% confidence interval, 1.38–11.50; *p* = 0.014). Otherwise, sex, age, or smoking status were not found to be favorable factors for T790M detection using plasma samples.

## 4. Discussion

This study showed that successful detection of the T790M mutation using plasma samples in patients with disease progression following first-line treatment with the first- or second-generation EGFR-TKI was associated with the tumor burden, including the number of metastatic organs. Although liquid biopsy using plasma samples for the detection of the T790M mutation is recommended initially according to the National Comprehensive Cancer Network guidelines [12], 40% of the patients with one or two metastatic organs at the time of re-biopsy had false negative results in the present study. Moreover, three or more metastatic sites at the initial diagnosis were identified as a favorable factor for the detection of the T790M mutation using plasma samples. To the best of our knowledge, this is the first report suggesting that the clinical phenotype is favorable to a liquid biopsy using plasma samples in detecting T790M mutations.

Re-biopsy using tissue samples for the detection of the T790M mutation was previously regarded as a gold standard, with high sensitivity and specificity [13,14]. However, there have been concerns about the safety of tissue re-biopsy. Nosaki et al. reported that the adverse event rate of re-biopsy in NSCLC patients was 5.8%, which was significantly higher than 1.3% of the initial biopsy [14]. Another limitation of tissue re-biopsy is that possible spatial heterogeneity cannot be completely overcome [11]. In addition, comparing with the initial tumor size, the size of the lesion may be much reduced due to the previous response of the first-line EGFR-TKI, which makes the tissue re-biopsy impossible [10,15,16,17]. Likewise, Uozu et al. reported that only 55% of patients who needed an identification of the T790M mutation were found to be feasible for tissue re-biopsy [10]. Because of these limitations, liquid biopsy using plasma samples has become new gold standard to detect the T790M mutation. However, the detection rate of T790M mutations using plasma samples was suboptimal, with a 67% sensitivity [16]. Therefore, it is important to find clinical conditions in which a liquid biopsy is advantageous for detecting the T790M mutation.

In our work, the detection rate of the T790M mutation using plasma samples was significantly increased in patients with a higher tumor burden, such as those with an advanced stage at the initial diagnosis and those with three or more metastatic organs at the time of the initial diagnosis or re-biopsy. This is the same as the research result of Fujii and colleagues; when the tumor burden is expanded in patients with advanced disease, DNA fragments are shedding into the bloodstream, which can be seen to increase the sensitivity of the liquid biopsy [17]. In a study by Sueoka-Aragane et al. on the relationship between plasma DNA and tumor status using an animal model, a progression in tumor status, such as the tumor volume or metastasis site, was related to an increase in tumor-derived DNA in the systemic circulation [18]. Therefore, it is reasonable to use a liquid biopsy using plasma samples if the tumor burden is relatively high according to the clinical factors such as the number of metastatic organs. Ulivi et al. conducted a practical study that suggests that a liquid biopsy is useful for predicting the disease prognosis and treatment response in patients with advanced or disseminated disease. [19] On the contrary, our results suggest that a tissue biopsy might be more useful in patients with few metastatic sites.

The present study has some limitations. First, the sample size was relatively small, and there is a difference in the number of subjects belonging to the two study groups. Second, there might be an inherent bias when statistically analyzing data with a retrospective study design. Third, tissue re-biopsy was not performed in all patients whose EGFR T790M mutation could not be detected by a liquid biopsy. When the EGFR T790M mutation is not detected by a plasma sample, EGFR mutation testing using tissue samples has traditionally been recommended by several guidelines [12,20,21,22]. However, in this study, due to the location of the lesion or the general condition of the patient, it was not possible to perform a tissue re-biopsy in all patients with negative plasma EGFR T790M results. Considering the issues presented above, a multicenter, large cohort, prospective study is required in the near future.

## 5. Conclusions

In conclusion, our results demonstrated that the detection rate of the EGFR T790M mutation using plasma samples was related to the tumor burden, especially the number of metastatic organs. Patients with three or more metastatic organs at the initial diagnosis might have a higher chance for the detection of the EGFR T790M mutation using plasma samples, whereas a tissue re-biopsy could be more diagnostic in patients with one or two metastatic organs.

## Figures and Tables

**Figure 1 cancers-15-01445-f001:**
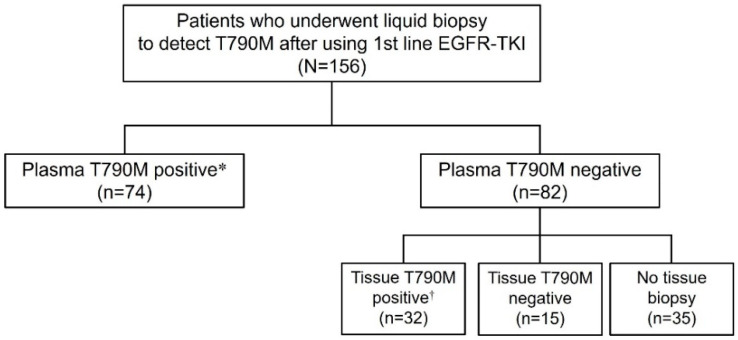
Flow diagram of study subjects. ***** Plasma positive group, ^†^ Plasma false negative group.

**Table 1 cancers-15-01445-t001:** Baseline characteristics of study subjects.

	Overall Group (N = 106)	Plasma Positive Group (n = 74)	Plasma False Negative Group (n = 32)	*p*-Value
Age, years	67 ± 10	68 ± 9	65 ± 13	0.191
Male gender	57 (54)	37 (50)	20 (63)	0.331
ECOG performance status				0.768
0	60 (56)	42 (57)	18 (56)	
1	42 (40)	28 (38)	14 (44)	
≥2	4 (4)	4 (5)	0 (0)	
Smoking status				0.626
Never smoker	65 (61)	47 (64)	18 (56)	
Ever smoker	41 (39)	27 (36)	14 (44)	
EGFR mutation status				0.601
19del	62 (58)	45 (60)	17 (53)	
L858R	44 (42)	29 (40)	15 (47)	
Initial stage *				0.001
I	5 (5)	3 (4)	2 (6)	
II	2 (2)	2 (3)	0 (0)	
III	15 (14)	7 (9)	8 (25)	
IV	84 (79)	62 (84)	22 (69)	

ECOG = Eastern Cooperative Oncology Group. The data were presented with mean ± standard deviation or No (%). * Based on the eighth edition of the American Joint Commission on Cancer TNM staging system.

**Table 2 cancers-15-01445-t002:** EGFR mutation status on secondary biopsy.

	Plasma Positive Group (n = 74)		Plasma False Negative Group (n = 32)
19del + T790M	46 (47)	Wild type	18 (28)
L858R + T790M	30 (31)	19del + T790M	17 (27)
19del	9 (9)	L858R + T790M	15 (23)
L858R	6 (6)	19del	8 (13)
T790M	4 (4)	L858R	6 (9)
Wild type	1 (1)		
9del + L858R + T790M	1 (1)		
L861Q + T790M	1 (1)		

RT-PCR = real time-PCR, PNA clamping = peptide nucleic acid clamping. The data were presented with No (%).

**Table 3 cancers-15-01445-t003:** Treatment-related parameters.

	Plasma Positive Group (n = 74)	Plasma False Negative Group (n = 32)	*p*-Value
Initial TKI			0.361
Gefitinib	22 (29)	7 (22)	
Erlotinib	10 (14)	2 (6)	
Afatinib	42 (57)	23 (72)	
Duration of first TKI, months	19 ± 14	24 ± 22	0.179
First response to initial TKI *			0.834
Partial response	42 (57)	19 (59)	
Stable disease	32 (43)	13 (41)	
Best response to initial TKI *			0.440
Partial response	43 (58)	19 (59)	
Stable disease	31 (42)	13 (41)	

TKI = tyrosine kinase inhibitor. The data were presented with mean ± standard deviation or No (%). * Treatment responses were analyzed according to response evaluation criteria in solid tumor (RECIST 1.1).

**Table 4 cancers-15-01445-t004:** TNM, clinical stage, and the number of metastatic organs at the time of re-biopsy for the detection of EGFR T790M mutation *.

	Plasma Positive Group (n = 74)	Plasma False Negative Group (n = 32)	*p*-Value
T stage			0.630
0	6 (8)	3 (9)	
1	14 (19)	5 (16)	
2	21 (28)	10 (31)	
3	7 (10)	6 (19)	
4	26 (35)	8 (25)	
N stage			0.685
0	18 (24)	7 (22)	
1	8 (11)	4 (12)	
2	19 (26)	5 (16)	
3	29 (39)	16 (50)	
M stage			0.239
0	4 (5)	4 (13)	
1	70 (95)	28 (87)	
Clinical stage			0.239
III	4 (5)	4 (13)	
IV	70 (95)	28 (87)	
Number of metastatic organs			0.028
1	8 (11)	2 (6)	
2	15 (20)	10 (30)	
3	24 (32)	13 (38)	
≥4	27 (37)	5 (15)	

* Based on the eighth edition of the American Joint Commission on Cancer TNM staging system.

**Table 5 cancers-15-01445-t005:** Univariate and multivariate analysis of plasma positive group.

Variables	Univariate Analysis		Multivariate Analysis	
	OR (95% CI)	*p*-Value	OR (95% CI)	*p*-Value
Sex, female	0.60 (0.25–1.39)	0.238	0.67 (0.24–1.81)	0.427
Age, years	1.03 (0.99–1.08)	0.135	1.03 (0.99–1.08)	0.197
Smoking status, ever smoker	0.74 (0.32–1.73)	0.482		
Pleural effusion, yes	0.66 (0.28–1.58)	0.342		
Histologic type, non-squamous	0.76 (0.04–6.23)	0.818		
Metastatic organs ≥ 3 at initial diagnosis, yes	3.30 (1.28–9.72)	0.019	3.71 (1.38–11.50)	0.014
Metastatic organs ≥ 3 at re-biopsy, yes	1.72 (0.73–4.06)	0.211	1.97 (0.80–4.87)	0.139

OR = odds ratio, CI = confidence interval.

## Data Availability

Data are available on request.

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
