# Peer review of "Favorable Conditions for the Detection of EGFR T790M Mutation Using Plasma Sample in Patients with Non-Small-Cell Lung Cancer"

_cancers, 2023, doi:10.3390/cancers15051445_

Round 1
Reviewer 1 Report
Dear all,
The authors presented a report entitled « Favorable Conditions for the Detection of EGFR T790M Mutation Using Plasma Sample in Patients with Non-Small Cell 3 Lung Cancer”. The work is interesting. They consolidate the fact that it is important to choose the more sensitive and specific technics for cfDNA molecular exploration. They have to include cfDNA concentration in their analysis since it is well-known that it is associated with the stage of cancer. Please download the manuscript, in which you will find some additional comments.

Author Response
We thank the reviewer for bringing this issue to attention. I have reviewed the highlighted part in the attached file and added the content as follows.
In patients with advanced lung cancer, the presence or absence of the T790M mutation has significant clinical implications for both patients and medical staff. Therefore, we added the content as follows (page 2, line 59-65):
“However, in patients in whom the T790M mutation was not detected, progression-free survival was only 3.1 months [9]. Thus, significant efforts must be made to detect T790M mutations in patients whose disease has progressed after treatment with 1st or 2nd generation EGFR-TKI. Therefore, verification of the presence or absence of T790M mutation has been considered as the most important process for determining the next treatment strategy after confirming the resistance to the first line EGFR-TKI treatment.”

Reviewer 2 Report
The manuscript focuses on the evaluation of EGFR T790M molecular alterations in a multicelnter retrospective ring trial represents a timely relevant manuscript where major integrations should be implemented to accept this paper for the publication
- In the introduction section, please, could the authors better discuss about the role of T790M in the clinical management of NSCLC patients?
- In the study design section, please, could the authors review the inclusion criteria for the selection of enrolled patients? In my opinion, the authors should betetr describe clinical and molecular characteristics of patient cohort population
- In the results section, I would suggest to include mutant allele fraction by using technical parameters of adopted technology to quantify detected aleterations
Author Response
â–ª Comment 1
In the introduction section, please, could the authors better discuss about the role of T790M in the clinical management of NSCLC patients?
â–ª Response
We thank the reviewer for bringing this issue to attention. In patients with advanced lung cancer, the presence or absence of the T790M mutation has significant clinical implications for both patients and medical staff. Therefore, the clinical benefit according to the presence or absence of T790M is further described as follows (see page 2, line 59-62):
“However, in patients in whom the T790M mutation was not detected, progression-free survival was only 3.1 months [9]. Thus, significant efforts must be made to detect T790M mutations in patients whose disease has progressed after treatment with 1st or 2nd generation EGFR-TKI.”
â–ª Comment 2
In the study design section, please, could the authors review the inclusion criteria for the selection of enrolled patients? In my opinion, the authors should better describe clinical and molecular characteristics of patient cohort population
â–ª Response
We thank the reviewer for bringing this issue to attention. The inclusion criteria for the enrolled patients have been described in detail, the patient's molecular profile has been added to the baseline characteristics of the results, and the same items have been added to the table. In addition, the clinical parameters have been described. Therefore, the added baseline characteristics are described as follows (see page 2, line 83-89, page 4, line 139-148 and table 1):
“The inclusion criteria for this study were a pathologically confirmed diagnosis of ad-vanced NSCLC, the presence of an EGFR exon 19 or to 21 mutation at initial diagnosis, disease progression after first-line systemic therapy with 1st- or 2nd-generation EGFR-TKIs (erlotinib, gefitinib, or afatinib), and availability of at least one liquid biopsy using plasma and/or tissue re-biopsy at the time of disease progression. Patients who had an exon 20 or T790M mutation at first diagnosis, or without a liquid biopsy after progression were excluded.”
“The Eastern Cooperative Oncology Group performance status was 0 for 42 individuals (57%), 1 for 28 individuals (38%), and ≥2 for 4 individuals (5%) in the plasma positive group. There was no significant difference between the two groups when compared to the plasma false-negative group (P = 0.768). The number of never-smokers was 47 (64%) in the plasma positive group and 18 (56%) in the plasma false-negative group; no significant difference was observed between the two groups (P = 0.626). Regarding EGFR mutation status, 45 (60%) and 17 (53%) patients had the 19del mutation and 29 (40%) and 15 (47%) patients had the L858R mutation in the plasma positive and false-negative groups, respectively. No significant difference was noted between the two groups.”
â–ª Comment 3
In the results section, I would suggest to include mutant allele fraction by using technical parameters of adopted technology to quantify detected alterations
â–ª Response
We thank the reviewer for bringing this issue to attention. In the Results section, we have added a description of the EGFR mutation detection findings and presented the data in Table 2, as follows (see page 5, line 158-164 and Table 2):
“3.2. EGFR mutation status on secondary biopsy
Real time-PCR (RT-PCR) and PNA clamping were used to detect EGFR mutations in secondary biopsies. In EGFR mutation tests using tissue and plasma in the plasma-positive group, 46 cases in 42 patients (47%) had the 19del+T790M mutation and 30 cases in 26 patients (31%) had the L858R+T790M mutation. In the plasma false-negative group, 18 patients (28%) had the wild type, 17 patients (27%) had the 19del+T790M mutation and 15 patients (23%) had the L858R+T790M mutation. (Table 2).”

Reviewer 3 Report
This study was concerned with the accuracy of detecting T790M mutations in patients with NSCLC treated with EGFR-TKI using plasma samples. The results showed that "64% of the patients with one or two metastatic organs at the time of re-biopsy had false negative results with plasma samples", while "the detection rate of T790M mutation using plasma sample was significantly increased in patients with more tumor burden". It was concluded that the detection rate of T790M mutation using plasma samples was positively related to the to the number of metastatic organs. There were similar studies published recently with similar observations [Biomedicines. 2021 Sep 23;9(10):1299. European Respiratory Journal 2018 52: PA1757]. This study did not office much new insight into how to improve the accuracy of mutant EGFR detection using plasma samples.
Line 21: Since it was indicated in Line 18 that 64% of patients with one or two metastatic organs had false negative results, the percent detection of the T790M mutation in patients with three or more metastatic organs should be provided.
In Table 1, it showed that 69% of patients in the plasma false negative group had stage IV NSCLC, which was much higher than patients with stage I, II and III NSCLC. This result was somewhat contradicted to the conclusion that "the detection rate of T790M mutation using plasma sample was significantly increased in patients with more tumor burden (Line 21-22)".
In Table 3, the total number of subjects with one or more metastatic organs (N = 70 for the positive group and N = 35 for the negative group) was different from the total number of subjects in the plasma positive group (N = 74) and in the plasma negative (N =32). Please explain.
Did tissue biopsies were conducted in patients in the plasma positive group to confirm that the plasma positive was true positive?
Author Response
â–ª Comment 1
This study was concerned with the accuracy of detecting T790M mutations in patients with NSCLC treated with EGFR-TKI using plasma samples. The results showed that "64% of the patients with one or two metastatic organs at the time of re-biopsy had false negative results with plasma samples", while "the detection rate of T790M mutation using plasma sample was significantly increased in patients with more tumor burden". It was concluded that the detection rate of T790M mutation using plasma samples was positively related to the to the number of metastatic organs. There were similar studies published recently with similar observations [Biomedicines. 2021 Sep 23;9(10):1299. European Respiratory Journal 2018 52: PA1757]. This study did not office much new insight into how to improve the accuracy of mutant EGFR detection using plasma samples.
â–ª Response
We thank the reviewer for providing valuable feedback regarding he clinical findings. We reviewed the presented data and, as you mentioned, disease progression, as assessed by liquid biopsy, appears to correlate closely with prognosis. The data presented above are easy to understand because they are presented clearly. However, our study differed in that it included the attached multivariate analysis data. Therefore, we refer to the above data and have added the following sentence to page 9, line 264-266:
“Ulivi et al. conducted a practical study that suggests that liquid biopsy is useful for predicting the disease prognosis and treatment response in patients with advanced or disseminated disease [19].”
â–ª Comment 2
Line 21: Since it was indicated in Line 18 that 64% of patients with one or two metastatic organs had false negative results, the percent detection of the T790M mutation in patients with three or more metastatic organs should be provided.
â–ª Response
We thank the reviewer for bringing this issue to attention. There were some errors in transcribing the calculation results, and the fraction occupied by one or two metastatic organs in the false-negative plasma group was corrected to 40% instead of 64%. The proportion of patients with three or more metastatic organs in the plasma positive group was 69%. We apologize for the error. The above content was added to the text as follows (see page 1, line 18-20, 33-35 and page 6, line 191-192):
“In this study, 40% of patients with one or two metastatic organs at the time of re-biopsy had false-negative plasma sample results, while 69% of patients with three or more metastatic organs at the time of re-biopsy had positive plasma results.”
“As a result, 40% of patients with one or two metastatic organs at the time of re-biopsy had false-negative plasma sample results, and 69% of patients with three or more metastatic organs at the time of re-biopsy had positive plasma results.”
“40% of the patients with one or two metastatic organs at the time of re-biopsy had false negative results in the present study.”
â–ª Comment 3
In Table 1, it showed that 69% of patients in the plasma false negative group had stage IV NSCLC, which was much higher than patients with stage I, II and III NSCLC. This result was somewhat contradicted to the conclusion that "the detection rate of T790M mutation using plasma sample was significantly increased in patients with more tumor burden (Line 21-22)".
â–ª Response
We thank the reviewer for bringing this issue to attention. As you pointed out, the proportion of stage 4 at the time of first diagnosis in the plasma false-negative group was relatively high (69%). However, the proportion of stage 4 in the plasma positive group was higher (84%), indicating a significant difference between the two groups. This result was due in part to the relatively small number of subjects in the plasma false-negative group compared to the plasma positive group. In summary, although there were many patients with advanced stage disease in the plasma false-negative group, there was no statistical significance when compared with the entire group; in the plasma-positive group, both the number and fraction of stage 4 patients were high,
â–ª Comment 4
In Table 3, the total number of subjects with one or more metastatic organs (N = 70 for the positive group and N = 35 for the negative group) was different from the total number of subjects in the plasma positive group (N = 74) and in the plasma negative (N =32). Please explain.
â–ª Response
We thank the reviewer for bringing this issue to attention. As you have pointed out, there was an error in the notation of the numbers in Table 3. This seems to be because the data were not updated in the previous table. We apologize for these errors in the table. Therefore, we have rechecked the data and created a revised table. Two subjects of false-negative plasma group were excluded because of disease progression due to an increase in the size of the main lesion without an increase in the number of metastatic organs. Therefore, 30 false-negative plasma subjects were included in this category. We changed the numbers in Table 4 (page 7).
â–ª Comment 5
Did tissue biopsies were conducted in patients in the plasma positive group to confirm that the plasma positive was true positive?
â–ª Response
As you mentioned, it is most desirable to detect EGFR in the tissue and plasma. However, when a liquid biopsy is performed, it is often impossible to acquire tissue owing to the deterioration of the patient’s general condition. Specifically, liquid biopsy for detecting the T790M mutation is crucial in cases of advanced EGFR-mutant non-small cell lung cancer (NSCLC) after treatment with first- or second-generation EGFR-TKIs, where the tumor tissue is either unavailable or limited [1]. In addition, both of the liquid biopsy methods used in this study were approved by the FDA for detecting EGFR mutations, so there was no difficulty in using them in clinical practice. Therefore, if the EGFR mutation test using plasma sample was positive, treatment was initiated by citing the EGFR test result without the need for tissue test confirmation.
References
- Ann Oncol. 2016;27(suppl 5):v1–v27.

Round 2
Reviewer 1 Report
The revised manuscript is acceptable for publication
Reviewer 2 Report
The manuscript may be accepted in the present form
Reviewer 3 Report
The authors have addressed my questions.